# Tacrolimus-Based Immunosuppressive Therapy Influences Sex Hormone Profile in Renal-Transplant Recipients—A Research Study

**DOI:** 10.3390/biology10080709

**Published:** 2021-07-24

**Authors:** Dagmara Szypulska-Koziarska, Aleksandra Wilk, Małgorzata Marchelek-Myśliwiec, Daria Śleboda-Taront, Barbara Wiszniewska

**Affiliations:** 1Department of Histology and Embryology, Pomeranian Medical University, Powst. Wlkp. 72, 70-111 Szczecin, Poland; aleksandra.wilk@pum.edu.pl (A.W.); barbara.wiszniewska@pum.edu.pl (B.W.); 2Department of Nephrology, Transplantology and Internal Medicine, Pomeranian Medical University, Powst. Wlkp. 72, 70-111 Szczecin, Poland; malgorzata.marchelek@gmail.com; 3Department of Laboratory Medicine, Pomeranian Medical University, Powst. Wlkp. 72, 70-111 Szczecin, Poland; d.sleboda@wp.pl

**Keywords:** hormones, immunosuppressants, renal transplantation

## Abstract

**Simple Summary:**

Although renal-transplant-recipients can lead much more comfortable life in comparison to patients on dialysis, they need to face other problems. One of them is lifetime immunosuppressive therapy on daily basis. Immunosuppressive regimen contains usually three different drugs and although each one of them is crucial to keep graft in good condition and to suppress immune response against the transplanted organ, they influence, among the other, reproductive system. In current paper we have observed that immunosuppressive therapy based on tacrolimus significantly affected the hormonal balance of sex hormones in both men and women. It is of great importance, as nowadays infertility is rising problem even in health people, therefore more attention should be paid to choose the best suitable immunosuppressive regiment for recipient in reproductive age.

**Abstract:**

It is estimated that approximately 20% of couples suffer from infertility worldwide and within renal-transplant recipients, this problem is 10 times more common. An intake of immunosuppressants may lead to hormonal imbalance. The aim of the study was to investigate the influence of tacrolimus-based therapy on the hormonal status of grafted patients. Blood samples were obtained from patients from the Department of Nephrology, Transplantology, and Internal Medicine of Independent Public Clinical Hospital No. 2, Pomeranian Medical University. All 121 patients had stable graft function for over 6 months. The blood plasma concentrations of luteinizing hormone, follicle-stimulating hormone, prolactin, testosterone, estradiol, cortisol were assessed by the electrochemiluminescence method. We observed decreased levels of prolactin (11.9 ng/mL) and cortisol (87.4 μg/mL) in patients under tacrolimus-based therapy. Tacrolimus-based therapy was also associated with increased testosterone and follicle-stimulating hormone in males, 4.04 ng/mL and 6.9 mLU/mL, respectively, and decreased testosterone levels in females, 0.121 ng/mL. We also assessed that immunosuppressive therapy based on tacrolimus is less nephrotoxic in comparison to other regimens. Concluding, tacrolimus-based therapy may influence the hormonal status of transplant recipients in the current study. Results presented here are believed to be helpful for clinicians and patients, especially within the aspect of willingness for biological offspring.

## 1. Introduction

It is currently estimated that approximately 20% of healthy couples suffer from infertility worldwide and within female renal-transplant recipients (RTR) pregnancy is even 10 times less common [1]. Among the patients with chronic kidney disease (CKD), frequent disturbances of the function of numerous endocrine glands are observed, which results from the impairment within the hypothalamic–pituitary–gonadal axis (HPGA) [1,2]. Disturbances in HPGA manifest differently in both sexes. In 70% of CKD, women ovarian cycles are disrupted, the luteal phase is shortened, anovulatory cycles are observed and, the appearance of menopause is diagnosed approximately 4.5 years earlier than in the case of healthy women [3]. The issue of male andropause is controversial since there is still no consensus between scientists on whether it exists or not. Nevertheless, in the latest European Male Aging Study and American Massachusetts Male Aging Study, it was indicated that male hypogonadotropic hypogonadism, termed also as andropause, becomes noticeable in men after 40 years of age and deepens with every passing year [4]. Disorders in HPGA among men may result in erectile dysfunctions, lowered libido, improper spermato- and steroidogenesis [5]. It is shown that decreased testosterone (T) level is associated with a higher risk of premature cardiovascular disease development, graft loss and mortality [6,7].

Numerous data suggest that the normal levels of sex hormones are restored 6 months after successful kidney transplantation (KT) [5,8,9]. The RTRs need to undergo immunosuppressive therapy (IT) daily, which may influence hormonal homeostasis.

For renal transplant recipients, an intake of immunosuppressive drugs (ID) is of utmost importance, since they prolong the vitality of the graft extending life expectancy. As kidneys exhibit great immunity, in transplant recipients, the immunosuppressive therapy most commonly includes three IDs. Nowadays in most transplant units, there are standards when choosing ID, however, the reason for the selection of particular drugs to be applied for an individual patient is based on the carefully carried out prior multifactorial analysis [6].

Tacrolimus (TAC) and cyclosporine A (CsA), belong to calcineurin inhibitors (CNIs). The other groups are: mTOR inhibitors with everolimus and sirolimus, inhibitors of cell divisions including inhibitors of inosine-5-monophosphate dehydrogenase (IMPDH) including mycophenolate mofetil (MMF) or sodium mycophenolate (MMS) and antimetabolite–azathioprine (AZA). The last group is glucocorticosteroids, most frequently prednisone (PRED) [10,11,12]. In female RTR in reproductive age, TAC in a multidrug regimen with PRED and MMF is associated with decreased prolactin (PRL) and luteinizing hormone (LH) concentrations and concomitantly increased estradiol (E2) and follicle-stimulating hormone levels [13]. On the contrary, in liver grafted patients under TAC and PRED-based therapy, a significantly increased level of PRL was observed even 13 months after transplantation (Tx) [14]. There are data suggesting that multidrug IT based on TAC with PRED or MMF with PRED and CsA used by young male RTR resulted in an elevated level of 17-hydroxyprogesteron. The same regimen in female RTR was associated with increased concentration of PRL and decreased level of T and dehydroepiandrosterone sulfate [15]. Male cardiac recipients under long-term IT based on TAC with MMF or MMF with CsA suffered from elevated concentrations of LH and FSH, concomitantly they complained about a reduction in T level [16]. Prednisone is widely used in recipients, especially short after Tx in order to avoid graft rejection. However, due to numerous side effects which appear to be associated with PRED usage, the dosage is usually gradually reduced. In Poland, doses of PRED in recipients are reduced or increased depending on numerous factors. These are, among the other, time after Tx, comorbidities including autoimmunological diseases, current graft function reflected by creatinine level, glycemia, etc. Little is known about the effect of IDs on hormonal status. However, histological and biochemical experiments conducted on male rats and pregnant female rats revealed that multidrug IT, in various regimens, significantly affected the liver and kidney [17,18,19]. There are data showing that CNIs inhibit the growth and steroidogenic capacity of Leydig cells, which are responsible for, among the other, T synthesis [6,20]. Tacrolimus was indicated to decrease T concentration in male RTR in the long-term IT [6]. Although TAC used to be considered as a safe drug for patients in case of desired pregnancy, nowadays, its intake is controversial and therefore more studies are required on this topic.

Apart from IDs, there are other environmental factors affecting human health, including hormonal balance. There is evidence that moderate wine consumption may be advantageous [21]. On the other hand, some data indicate that alcohol intake impairs hormonal balance or increases the risk of breast cancer development [22].

It is believed that after successful KT, an impairment in HPGA may be recovered, however, it is still unclear to what extent it proceeds. According to the literature, it is impossible to predict the future gonadal function in RTR [5]. The effects of the intake of TAC are still controversial, therefore the aim of the current study was to investigate the influence of multidrug IT based on TAC on the concentrations of sex hormones profile in serum of RTR, including: LH, FSH, PRL, T, E2, cortisol (CORT) by sex, age and codrugs likewise MMF, PRED.

## 2. Patients and Methods

### 2.1. Study Design and Population

The current study was approved by the Bioethics Committee of the Pomeranian Medical University (decision KB-0012/74/17). The research was carried out from 2017 to 2019. One hundred and nineteen (121) patients of the Department of Nephrology, Transplantology, and Internal Medicine of Independent Public Clinical Hospital No. 2, Pomeranian Medical University, in the city of Szczecin in northwestern Poland who underwent kidney transplantation were enrolled into the study. A total of 57 women aged 24 to 71 years and 64 men 24 to 71 years of age, all of them with stable for the minimum 6 months graft function (the average duration of graft functioning was assessed as 8 years) were included in this trial. All the patients were presented with detailed information about the research, including the purposes and subsequent steps of the study. Before the blood was collected, each patient submitted voluntary consent for participation. All the premenopausal women were in the follicular phase of the menstruation cycle when donating the blood. in A 10-cc blood sample was obtained in the morning (7.30. am–9.30. am) by a well-qualified nurse from the elbow vein during the diagnostic workup and collected into certified tubes by a vacuum method (Vacutainer System, royal blue cap). Blood samples were centrifuged within 30 and 120 min of collection to separate the serum from the cellular fraction. The serum samples were stored at −80 °C until the FSH, LH, T, E2, PRL and CORT assays were performed. The biochemical analyzes of the concentration of the abovementioned hormones were performed with the usage of the electrochemiluminescence method. All methods applied in the current study were performed in accordance with the relevant guidelines and regulations.

In the current study the patients were divided into two groups: (i) TAC+, patients whose multidrug IT, administered orally, was based on tacrolimus; (ii) control group of patients (TAC-) whose multidrug IT, administered orally, excluded TAC but contained instead different drugs in various combination, likewise sirolimus, everolimus, CsA, MMS, AZA or PRED. The dosage of all the IDs used by patients was different for each patient depending on time after Tx, body weight, the concentration of particular drug in the blood, the lifespan and graft function. A total of 26 patients (21.8%) suffered from diabetes mellitus in mild form.

All the demographical data concerning patients are presented in Table 1. and Table 2. The data concerning percentage of renal-transplant recipients receiving following drugs in multidrug regimens is presented in Figure 1.

In current study all the renal-transplant recipients were administered multidrug immunosuppressive therapy. Presented below is percentage of patients taking a given drug.

### 2.2. Hormones Profile Assay

The concentrations of FSH, LH, T, E2, PRL and CORT were examined with the application of the electrochemiluminescence method (ECLIA), using ROCHE reagent kits suitable for a particular hormone, accordingly to the manual. All of the biochemical analyses were performed on a Cobas 8000 analyzer (Cobas 8000, e-801).

Based on the available manual, the following steps were performed for assessing the concentration of a particular hormone:Incubation of 30 μL of the sample with biotinylated monoclonal antibodies, specific for particular hormone, and with monoclonal antibodies specific for particular hormone labeled with a ruthenium complex.Addition of streptavidin-coated microparticles to the obtained complex.Transfer of the reaction mixture to the measuring chamber.Removal of unbound substances.Measurement of the electrochemiluminescence reactions of the obtained complexes and the photon emission excited by the voltage applied to the electrode by a photomultiplier.Reading of the results from the calibration curve.

### 2.3. Statistical Analysis

The values of the quantitative variables were compared between groups using the nonparametric Mann–Whitney *U*-test, as the data distribution was not consistent with the expected normal distribution. Therefore, median (Med), lower quartile (Q1) and upper quartile (Q3) were established for the concentrations of FSH, LH, T, E2, PRL and CORT. Moreover, a relationship between hormone profile and concentration of TAC was analyzed using a linear regression method. In addition, Spearman’s rank correlation coefficients (r_s_) were determined. The cut-off level of statistical significance was set at *p* < 0.05. The statistical analysis employed StatSoft Statistica 13.3 software and Microsoft Excel 2015. The obtained results are presented in Table 3, Table 4, Table 5, Table 6, Table 7 and Table 8 and in Figure 2.

## 3. Results

### 3.1. Serum Hormones Concentration in Renal Transplant Recipients by Sex

In female patients subjected to the IT based on TAC we observed significantly lower concentrations of T (*p* < 0.05), PRL (*p* < 0.005) and CORT (*p* < 0.05), in comparison to TAC- patients. Additionally, FSH concentration was over 25% higher in TAC+ woman group (Table 3).

In male RTR, the concentrations of FSH and T were found to be significantly higher in TAC+ patients, comparing to TAC- patients, (*p* < 0.05) (Table 3).

### 3.2. Serum Hormones Concentration in Renal Transplant Recipients by Sex and Age

Furthermore, we divided the results obtained by age into the following groups; females over and under 45 years of age, as in kidney-grafted women menopause is approximately 4.5 years earlier than in healthy women; males over and under 45 years of age. We then considered the influence of TAC intake on hormones levels in serum.

#### 3.2.1. Serum Hormones Concentration in Female Renal Transplant Recipients by Age

Regarding the age of years of female patients, in the younger group, we found the concentration of CORT to be 30% lower (*p* < 0.05) in TAC+ patients, comparing to TAC- patients (Table 4).

In the older group of female patients, it was observed that the concentration of PRL was almost 35% lower in women under the TAC therapy when compared with TAC-female patients, *p* < 0.005 (Table 4).

#### 3.2.2. Serum Hormones Concentration in Male Renal Transplant Recipients by Age

Taking into account the concentration of CORT in the younger group of males, it was found out to be 20% lower in patients treated with TAC, comparing to TAC- patients, *p* < 0.05 (Table 5).

### 3.3. Hormone Profile under Co-Therapy

We additionally investigated the concentration of the hormone in TAC+ recipients in terms of codrug therapy intake. We examined the influence of PRED and MMF as codrugs for TAC, however, we did not find any statistically important differences (*p* > 0.05) (Figure 2).

### 3.4. Correlations between Hormones Concentration

The Spearman’s rank correlation coefficient showed numerous positive and negative synergistic relationships between analyzed hormones (Table 6).

### 3.5. The Relationship of Hormones Concentration with TAC Level by Sex

The linear regression did not show any significant relationship between analyzed hormone levels and TAC concentrations (Table 7).

### 3.6. The Relationship of TAC Concentration with Age

We performed statistical analyses regarding the relationship between TAC concentration and age in patients, however, we did not find any significant correlation of the abovementioned in female RTR nor in male RTR (data not shown).

### 3.7. The Kidney Parameters by TAC Intake

We assessed the levels of creatinine and GFR in two groups of patients; patients whose immunosuppressive regimen was based on TAC, and in the group of patients whose immunosuppressive regimen exclude TAC. We observed a statistically significant decreased level of creatinine and concomitantly increased GFR in TAC+ recipients in comparison to TAC- patients (Table 8).

## 4. Discussion

Maintaining hormonal balance is a complex issue, particularly in patients suffering from CKD [6]. To our best knowledge, there is no available literature concerning the influence of multidrug IT based on tacrolimus on the hormonal status of RTR. Mostly, researches are based on one drug association with HPGA [2,23,24,25]. Rising infertility seems to be a meaningful problem. Moreover, most of the obtainable studies regard the comparison between serum concentrations of hormones in dialysis patients *versus* the serum hormones level of RTR [7,8,26].

Although TAC and CsA act on the same molecules and belong to the same group of immunosuppressive drugs, they are indeed two different drugs. The properties and effects that they exert are different [10,11,12]. Nowadays, TAC is more commonly applied to renal transplant recipients, which can be observed in the current study, since patients whose immunosuppressive therapy is based on TAC constitute almost 74% of all of the graft recipients. Furthermore, it is of note that our data from the current study reflects the most reliable conditions of treatment of kidney transplant recipients. These patients use multidrug immunosuppressive therapy, based on three drugs, therefore we compared the hormonal balance TAC+ and TAC- patients, despite having known that TAC and CsA are both calcineurin inhibitors.

Abnormalities within the HPGA are the well-known clinical problems in CKD, and although after successful KT the levels of the hormone should normalize, an intake of ID can affect the concentration of hormones. To explore the current topic we divided obtained results by sex. We revealed significantly decreased levels of PRL in TAC+ women, comparing to TAC- women. Our results stay contrary to others, who observed no influence of ID on the concentration of PRL after KT [1,7,27]. On the other hand, most of the available literature does not regard the hormonal status exclusively in women. Our analysis seems to reflect the hormonal status in many aspects, which makes it more valuable and worthy of particular attention. Taking into account our results and available data, this topic seems to interesting and requires extensive and in-full depth researches.

Furthermore, we observed the trend of decreasing concentration of PRL of women above 45 years of age under TAC-based treatment. Unfortunately, little is known on the topic in reference to the age of patients, as available researches do not classify the patient according to their age [1,7,27].

Disturbance of any hormones in the organism may lead to problems with fertility. Cortisol is already known for antiphlogistic properties, however recent in vitro research revealed, that CORT upregulates the expression of genes of claudin-3, claudin-4, E-cadherin, zona occludin-1 and desmoglein-1, which are involved in the formation of tight-junction in the epithelium, among others, in the oviducts, that is crucial for fertility [28]. Currently, in all the TAC+ patients, we found a significantly lower level of cortisol, when compared with TAC- patients. So far, there is no available data regarding the influence of TAC-based therapy in RTR on the concentration of CORT in the aspect of fertility, therefore our research seems to be innovative. This aspect requires to be widened, especially taking into account, that according to the statistics, around 40% of male RTR are below 50 years of age. The aspect of female menopause seems to be well-known in opposite to male andropause, which is controversial due to its gradual characteristic. However, according to the scientists from European Male Aging Study and Massachusetts Male Aging Study, the concentration of total T and free T is decreasing by 1.6% and 2–3% per year, respectively [4]. This finding seems to be of great importance in the aspect of willingness for biological offspring [29]. That aspect, potentially, remains of interest to the clinicians who decide which regimen would be the best fit for patients of reproductive age. Results obtained in the current study suggest that TAC-based treatment may influence the homeostasis of CORT in both, male and female RTR patients.

The imbalance in the synthesis/secretion of FSH may result in impairment of spermatogenesis, therefore, affecting procreation [5]. Interestingly, in the current study in male patients, we observed a significantly increased level of FSH in TAC+ group, in comparison to TAC-. It is to be considered why the TAC does not influence the FSH level in a woman. Perhaps the influence that TAC exerts on biochemical parameters depends on sex. According to that issue, more broadened researches on the topic should be conducted. The data regarding the influence of TAC-based IT in case of the FSH level is hardly available. Hamdi et al. [5] found the concentration of FSH in RTR to be similar to our result. Of note, Hamdi did not divide patients depending on the type of regimen in the aspect of multidrug therapy, so the influence of TAC was not examined and explained widely. The data concerning mTORs inhibitors and their influence on FSH remain controversial. Sirolimus in co-therapy significantly decreases the concentration of FSH [30]. Another study showed that IT-based on mTOR inhibitors led to an increased FSH level in males [5]. Various regimens exhibit different mechanisms of action, though CNIs influence hormonal status in a different way, comparing to mTORs. Therefore, it remains controversial and unexplained whether KT and IT improve reproductive function or not.

Simultaneously with gonadotropins, testosterone performs an extremely important function in preserving the ability of procreation in both sexes [31,32]. Currently, we observed that in females on TAC-based IT the T level was significantly decreased. Interestingly, this trend in male patients was reversed. Studies performed by Fritsche et al. [30] and Huyghe et al. [33] showed that administration of mTOR inhibitors was associated with significantly decreased concentration of T in male RTR, in comparison to other regimens. Our results seem to be similar, however only in men. That may be related with the differences in the metabolism and the biochemical microenvironment that greatly differs between sexes.

Renal transplant recipients are usually supposed to obtain more than one ID simultaneously, due to their different mechanisms of action. Therefore, we performed a multivariate regression analysis to verify if these codrugs exert any impact on the examined parameters. However, we did not observe any significant influence in an aspect of co-therapy for TAC, on the concentration of any of examined hormones. As mentioned above, little is known on the topic, therefore a continuation of this aspect of our research seems to be meaningful.

In the current study, we examined a correlation co-efficiency between all of the analyzed hormones. Of note, we observed a positive synergistic correlation between FSH and LH in all examined groups of patients. Moreover, a positive correlation was clearly visible between E2 and PRL in TAC+ patients and within all patients. Surprisingly, we revealed numerous negative correlations between steroidal hormones and pituitary trophic hormones in the TAC+ group. This significance however was not observed in TAC- group. It seems possible, that TAC, being a very strong calcineurin inhibitor influence steroidogenesis. No literature in the aspect of co-efficiency of various hormones in RTR under multidrug IT is available, therefore the topic requires to be widened.

As calcineurin inhibitors, including TAC, are toxic for the nervous system it is of great importance to monitor its concentration in the blood of patients. In the current paper, we assessed also whether there is any relationship between TAC concentration and endocrine function, however, we did not find any of that in female nor male patients. Perhaps doses of TAC used by our patients, do not influence significantly the concentration of any of the sex-related hormones analyzed by us. Nevertheless, since there is a lack of data in the aspect of correlation of concentration of TAC and level of a particular hormone in kidney-grafted patients under multidrug IT, it seems to be necessary to widen the topic.

Additionally, we compared the obtained results of the concentration of hormones with the normal range. Taking into account the homeostasis of T, we revealed that it was greatly affected only in older female patients, whereas in males it remained within the reference values. In females above 45 years of age, we assessed the concentration of T to be more than 25 times higher above the upper limit of the norm. Interestingly, this enormously high level of T was evident only in TAC- patients. Therefore, it seems that TAC is a safe drug for all patients from the point of view of T homeostasis. Regarding the concentration of CORT and FSH, we observed no deviations from the norms in the aspect of TAC or any other ID intake. Perhaps, the homeostasis of these hormones is stable enough and does not subjacent to change under the IF.

Eventually, we assessed whether the intake of TAC in multidrug therapy influences the most basic parameters of kidney function, in comparison to other regimens excluding TAC. We found out that in patients whose immunosuppressive therapy was based on TAC the level of creatinine was significantly lower and concomitantly a GFR parameter was improved, in comparison to the same parameters in patients, whose regimens excluded TAC. It may be therefore concluded that TAC seems to be less nephrotoxic when compared with another immunosuppressive drug, at least in the current experimental setup.

We are aware of some weaknesses of the current study. Among others, small sample size is one of them. We also faced the problems in discussion of obtained results, since the data concerning the influence of IT based on tacrolimus is poor in the aspect of hormones regulating fertility.

To summarize, IT based on tacrolimus was associated with disturbed HPGA, which has been currently revealed. Nevertheless, modern IT is credited for the broadening of knowledge concerning the influence of a single drug and the interaction between all the drugs supplied by RTR, the pregnancy after KT seems to be no longer impossible. On the other hand, each pregnancy may overload the function of the graft. Based on the performed analysis, it can be concluded that immunosuppressive therapy based on TAC seems to be safer for the kidneys in comparison to other regimens. Thus, keeping in mind, that IT influences hormonal homeostasis, great attention should be paid to any single abnormality revealed during such pregnancy.

## 5. Conclusions

Based on the results we achieved, it can be concluded that TAC-based immunosuppressive therapy exerts an impact on HPGA, which manifests as significant changes in the concentration of PRL, CORT, FSH, E2 and T in blood.

## Figures and Tables

**Figure 1 biology-10-00709-f001:**
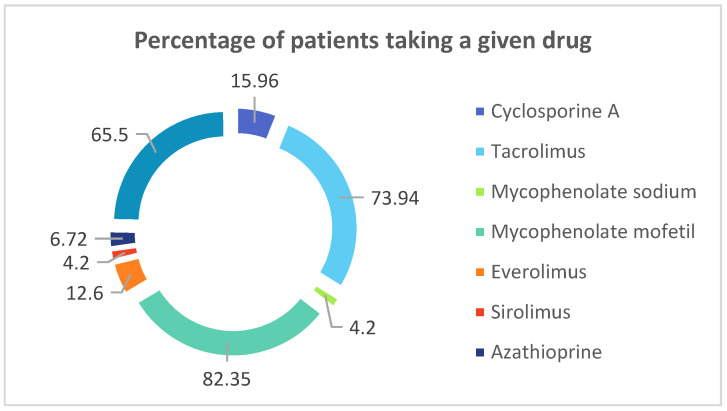
Percentage of renal-transplant recipients receiving following drugs in multidrug regimens.

**Figure 2 biology-10-00709-f002:**
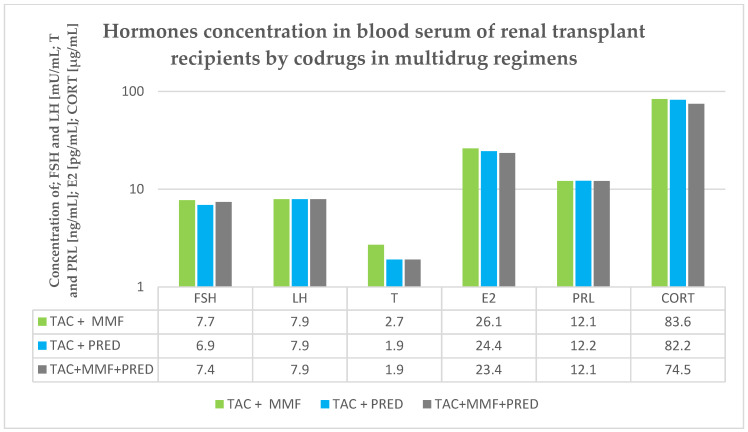
Hormones concentration in blood serum in renal transplant recipients by codrug in multidrug regimens.

**Table 1 biology-10-00709-t001:** Demographical and clinical data of patients—women.

Parameter		TAC+		TAC-	
		<45	>45	<45	>45
AGE [years]	n	19	21	7	10
	Med	39	62	37	61.5
	Q1–Q3	30–42	56–65	30–42	59–64
SMOKING	n	5	4	6	5
CONTRACEPTIVES	n	1	0	1	0
HRT	n	1	3	1	1
BMI [kg/m^2^]	n	19	21	7	10
	Med	23	23.6	22.6	24.05
	Q1–Q3	21.6–23.6	22.5–25.00	21.2–24.2	22.6–24.6
GFR [mL/min/m^3^]	n	19	21	7	10
	Med	55	51	54	33
	Q1–Q3	39.0–65.0	35.0–61.0	30.0–90.0	26.0–56.0
CRE [mg/dL]	n	19	21	7	10
	Med	1.22	1.2	1.28	1.75
	Q1–Q3	1.05–1.61	0.98–1.59	0.87–2.02	1.07–2.12
HEM [mmol/L]	n	18	21	7	7
	Med	7.95	8	7	7.6
	Q1–Q3	7.5–8.5	7.4–8.5	6.1–8.7	6.5–8.4
ALT [IU/L]	n	19	21	7	10
	Med	17	19	17	19
	Q1–Q3	14.0–27.0	14.0–25.0	10.0–21.0	15.0–20.0
PRED [mg/d]	n	3	1	3	2
	Med	5	5	6.6	5
	Q1–Q3	4.00–6.00	5.00–5.00	5.00–10.00	5.00–5.00
PRED [mg/kg]	n	3	1	3	2
	Med	0.08	0.08	0.09	0.08
	Q1–Q3	0.08–0.09	0.08–0.08	0.08–0.1	0.08–0.08

n: number of patients; Med: median; Q1: lower quartile; Q3- upper quartile; <45: patients below 45 years of age; >45: patients above 45 years of age HRT: hormonal replacement therapy; BMI: body mass index; GFR: glomerular filtration rate; CRE: creatinine; HEM: hemoglobin; ALT: alanine aminotransferase; PRED: prednisone.

**Table 2 biology-10-00709-t002:** Demographical and clinical data of patients—men.

Parameter		TAC+		TAC-	
		<45	>45	<45	>45
Age [years]	n	16	30	4	14
	Med	34.5	60	36	61.5
	Q1–Q3	33–38	55–63	33–38	56–64
SMOKING	n	5	7	2	2
ED	n	2	10	1	5
BMI [kg/m^2^]	n	16	30	4	14
	Med	23.6	23.6	22.55	23.95
	Q1–Q3	22.5–24.5	22.5–24.5	21.85–23.3	23.0–25.0
GFR [mL/min/m^3^]	n	16	30	4	14
	Med	69	53	53.5	39.5
	Q1–Q3	50.5–78.0	44.0–73.0	35.0–76.6	25.0–53.0
CRE [mg/dL]	n	16	30	4	14
	Med	1.33	1.34	1.63	1.76
	Q1–Q3	1.17–1.74	1.09–1.78	1.28–2.68	1.4–2.88
HEM [mmol/L]	n	16	30	4	14
	Med	8	19	8.6	8.1
	Q1–Q3	7.7–8.55	15.0–26.0	7.15–9.7	7.0–9.2
ALT [IU/L]	Q1–Q3	16	30	4	14
	Med	20.5	19	33.5	25
	Q1–Q3	14.5–29.5	15.0–26.0	23.5–37.0	15.0–29.0
PRED [mg/d]	n	1	2	1	2
	Med	5	7	5	7.5
	Q1–Q3	5.00–5.00	4.00–10.00	5.00–5.00	5.00–10.00
PRED [mg/kg]	n	1	2	1	2
	Med	0.08	0.08	0.07	0.09
	Q1–Q3	0.08–0.08	0.07–0.08	0.07–0.07	0.08–0.09

n: number of patients; Med: median; Q1: lower quartile; Q3- upper quartile; <45: patients below 45 years of age; >45: patients above 45 years of age; ED: erectile disorders; BMI: body mass index; GFR: glomerular filtration rate; CRE: creatinine; HEM: hemoglobin; ALT: alanine aminotransferase; PRED: prednisone.

**Table 3 biology-10-00709-t003:** Hormones and TAC concentrations in blood of female and male renal transplant recipients.

Parameter		Females (n = 57)		Males (n = 62)	
		TAC+ (n = 40)	TAC- (n = 17)	TAC+ (n = 44)	TAC- (n = 18)
FSH [mLU/mL]	Med Q1–Q3	21.50 3.75–80.50	8.40 4.20–28.10	6.90 4.35–10.45	4.80 3.61–7.30
		MW *U p* = 0.453		MW *U* * *p* = 0.048	
LH [mLU/mL]	Med Q1–Q3	26.75 8.46–50.80	14.80 9.71–21.90	7.10 4.15–9.65	7.26 5.60–13.30
		MW *U p* = 0.524		MW *U p* = 0.393	
T [ng/mL]	Med Q1–Q3	0.121 0.05–0.21	0.137 0.02–0.53	4.04 3.31–4.78	3.47 2.56–4.25
		MW *U* * *p* = 0.048		MW *U* * *p* = 0.042	
E2 [pg/mL]	Med Q1–Q3	32.90 8.30–137.00	55.40 6.13–136.80	26.55 18.45–30.25	24.45 17.40–31.40
		MW *U p* = 0.870		MW *U p* = 0.912	
PRL [ng/mL]	Med Q1–Q3	14.40 10.60–18.40	17.60 15.90–127.00	11.30 8.30–12.50	12.45 9.31–19.70
		MW *U* ** *p* = 0.004		MW *U p* = 0.172	
CORT [μg/mL]	Med Q1–Q3	82.45 57.55–115.00	105.00 88.40–165.00	82.60 55.70–116.00	97.80 77.30–123.00
		MW *U* * *p* = 0.011		MW *U p* = 0.300	
TAC[ng/mL]	Med Q1–Q3	6.45 4.90–8.20	-	5.70 4.70–7.90	-

n: number of patients; Med: median; Q1: lower quartile; Q3- upper quartile; TAC+: immunosuppressive therapy based on tacrolimus; TAC-: immunosuppressive therapy based on drugs excluding tacrolimus; MW *U:* Mann–Whitney *U*-test; * statistical significance with *p* < 0.05; ** statistical significance with *p* < 0.005

**Table 4 biology-10-00709-t004:** Hormone and TAC concentrations in blood of female renal transplant recipients under and over 45 years old.

Parameter		<45 (n = 26)		>45 (n = 31)	
		TAC+ (n = 19)	TAC- (n = 7)	TAC+ (n = 21)	TAC- (n = 10)
FSH [mLU/mL]	Med Q1–Q3	3.90 2.80–6.60	4.50 2.30–6.27	73.90 53.60–91.30	24.80 8.40–81.80
		MW *U p* = 0.917		MW *U p* = 0.156	
LH [mLU/mL]	Med Q1–Q3	9.60 4.80–20.90	9.80 5.59–17.20	45.70 36.00–69.70	18.60 10.70–79.80
		MW *U p* = 0.817		MW *U p* = 0.386	
T [ng/mL]	Med Q1–Q3	0.13 0.06–0.27	0.17 0.02–0.53	0.09 0.02–0.21	0.09 0.02–3.13
		MW *U p* = 0.602		MW *U p* = 0.945	
E2 [pg/mL]	Med Q1–Q3	132.0 57.50–179.00	58.20 4.50– 158.00	11.70 4.50–20.30	38.70 7.76–88.00
		MW *U p* = 0.272		MW *U p* = 0.056	
PRL [ng/mL]	Med Q1–Q3	16.40 12.80–23.10	17.00 14.10–19.60	11.45 9.12–16.10	20.65 16.00–32.90
		MW *U p* = 0.09		MW *U* * *p* = 0.007	
CORT [μg/mL]	Med Q1–Q3	75.10 45.10–124.00	104.00 88.40–179.00	97.30 61.50–110.00	105.00 87.50–165.00
		MW *U* * *p* = 0.049		MW *U p* = 0.122	
TAC[ng/mL]	Med Q1–Q3	6.80 5.30–8.20	-	6.40 4.70–8.00	-

n: number of patients; Med: median; Q1: lower quartile; Q3- upper quartile <45: patients below 45 years of age; >45: patients above 45 years of age; TAC+: immunosuppressive therapy based on tacrolimus; TAC-: immunosuppressive therapy based on drugs excluding tacrolimus; MW *U*: Mann–Whitney *U*-test; * statistical significance.

**Table 5 biology-10-00709-t005:** Hormone and TAC concentrations in blood of male renal transplant recipients under and over 45 years old.

Parameter		<45 (n = 21)		>45 (n = 41)	
		TAC+ (n = 16)	TAC- (n = 5)	TAC+ (n = 28)	TAC- (n = 13)
FSH [mLU/mL]	Med Q1–Q3	5.81 3.75–9.34	4.76 3.81–5.92	7.35 4.99–11.70	4.90 3.61–7.96
		MW *U p* = 0.363		MW *U p* = 0.207	
LH [mLU/mL]	Med Q1–Q3	5.73 3.89–7.60	7.30 6.40–12.72	7.35 4.66–10.40	7.22 5.08–13.30
		MW *U p* = 0.301		MW *U p* = 0.833	
T [ng/mL]	Med Q1–Q3	4.03 3.61–4.91	3.85 3.21–5.33	3.96 3.02–4.66	3.15 2.50–4.25
		MW *U p* = 0.649		MW *U p* = 0.244	
E2 [pg/mL]	Med Q1–Q3	20.10 13.20–26.00	18.70 10.19–19.80	27.80 21.65–33.90	29.10 22.50–38.60
		MW *U p* = 0.710		MW *U p* = 0.988	
PRL [ng/mL]	Med Q1–Q3	11.40 8.82–12.50	16.10 13.65–22.20	11.20 9.30–13.75	12.30 7.92–18.10
		MW *U p* = 0.052		MW *U p* = 0.726	
CORT [μg/mL]	Med Q1–Q3	69.25 51.75–85.0	85.00 60.65–111.0	103.0 61.5–124.0	98.3 83.7–123.0
		MW *U* * *p* = 0.046		MW *U p* = 0.862	
TAC[ng/mL]	Med Q1–Q3	6.90 5.00–9.50	-	5.60 4.70–7.70	-

n: number of patients; Med: median; Q1: lower quartile; Q3- upper quartile <45: patients below 45 years of age; >45: patients above 45 years of age; TAC+: immunosuppressive therapy based on tacrolimus; TAC-: immunosuppressive therapy based on drugs excluding tacrolimus; MW *U*: Mann–Whitney *U*-test; * statistical significance.

**Table 6 biology-10-00709-t006:** Correlation coefficients of FSH, LH, T, E2, PRL, CORT in grafted patients, by TAC intake.

Correlated Hormones	Correlation Coefficients		
	TAC+	TAC-	All Patients
**FSH/LH**	0.73 *	0.66 *	0.72 *
**FSH/E2**	−0.61 *	−0.16	−0.49 *
**FSH/PRL**	−0.11	0.24	−0.05
**FSH/T**	−0.18	−0.02	−0.13
**FSH/CORT**	0.07	0.02	0.002
**LH/E2**	−0.23 *	0.21	−0.10
**LH/PRL**	0.13	0.27	0.16
**LH/T**	−0.35 *	−0.04	−0.27 *
**LH/CORT**	−0.04	−0.09	−0.05
**E2/PRL**	0.30 *	0.20	0.28 *
**E2/T**	−0.01	0.05	0.01
**E2/CORT**	0.09	0.04	0.09
**PRL/T**	−0.29 *	0.01	−0.21 *
**PRL/CORT**	0.11	0.07	0.14
**T/CORT**	−0.02	0.06	0.005

TAC+: immunosuppressive therapy based on tacrolimus TAC-: immunosuppressive therapy based on drugs excluding tacrolimus; *: statistically significant difference with *p* < 0.05.

**Table 7 biology-10-00709-t007:** Linear regression of FSH, LH, T, E2, PRL, CORT and PRED in TAC+ grafted female and male patients, by concentration of TAC.

Hormone	Females (n = 40)			Males (n = 44)		
	Regression Equation	R	*p*	Regression Equation	R	*p*
H	Y = 38.22 + 1.97x	0.11	0.51	Y = 1.22 + 1.94x	0.22	0.164
LH	Y = 27.32 + 1.49x	0.12	0.46	Y = 4.42 + 1.02x	0.17	0.275
E2	Y = 137.41 − 8.42x	−0.17	0.32	Y = 31.32 − 0.86x	−0.23	0.148
PRL	Y = 13.87 + 0.20x	0.07	0.68	Y = 11.67 + 0.12x	0.04	0.771
T	Y = 0.17 − 0.004x	−0.07	0.69	Y = 4.87 − 0.11x	−0.15	0.333
CORT	Y = 98.81 − 0.84x	−0.06	0.73	Y = 126.96 − 5.69	−0.37	0.14

n: number of patients; r: Pearson correlation coefficient, *: statistically significant difference with *p* < 0.05.

**Table 8 biology-10-00709-t008:** The kidney parameters creatinine and GFR in patients by concentration of TAC.

Parameter		TAC+ (n = 84)	TAC- (n = 35)
GFR [mL/min/m^3^]	Med Q1–Q3	54.00 10.00–100.00	40.00 9.00–110.00
		MW *U p* = 0.018	
CREATININE [mg/dL]	Med Q1–Q3	1.285 0.72–4.42	1.57 0.74–5.92
		MW *U p* = 0.022	

n: number of patients; Med: median; Q1: lower quartile; Q3- upper quartile; TAC+: immunosuppressive therapy based on tacrolimus; TAC-: immunosuppressive therapy based on drugs excluding tacrolimus; MW *U*: Mann–Whitney *U*-test.

## Data Availability

All data for this paper can be found in the text.

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
