# Peer review of "Tacrolimus-Based Immunosuppressive Therapy Influences Sex Hormone Profile in Renal-Transplant Recipients—A Research Study"

_biology, 2021, doi:10.3390/biology10080709_

Round 1

Reviewer 1 Report

Szypulska et al performed a crosssectional study correlating immunosuppression and wine consumption with steroid- and non-steroid sex hormones in stable kidney transplant recipients. The authors hypothesized that in analogy to published in vivo data Tacrolimus-based immunosuppression reduces sex hormone production and circulatory levels in this patient cohort. The study consists of 119 kidney transplant recipients of both sexes, with a relatively large proportion of males and females aged 45 years and older. They quantified antropometric values, allograft function and steorid- and non-steorid sex levels in the plasma. Overall the authors claim that in female patients on Tac-based immunosuppression had a significantly lower level of Testosteron, Prolactin and Cortisol. In Males, incerased levels of Testerosteron and FSH were reported. Furthermore, the claim that Wine consumption (dichtomous variable) increases Cortisol levels.

Overall, while correlation of immunosuppression and social behaviour on reproductive function is attractive, we identify various conceptional, design and statistical problems in this manuscript.

- Cofounders: the study is ill controlled for relevant clinical and pre-analytic co-founders, including permenstrual state of female participants, comedication (including hormonal replacement therapy), auotimmune disorders and number of rejections, previous immunosuppression before and after transplantation, (Cyclophosphamide, Steroids, Induction Therapy, Chemotherapy), dialysis time and number of kidney and other solid organ transplants, prevalence of primary amenorrhea, interaction of hormone levels and eGFR.

- Pre-analysis: The study does not describe in the methods details about sampling: association with menstruation in premenopausal women, time of blood sampling,

- Reporting Bias: A significant part of the manuscript covers correlation of wine consumption on sex hormone levels, although the study uses wine consuption as a dichotomous variable. Clearly, self reporting alcohol consumption has limitations in such research and must be analysed as a continous variable.

- Text and References: The introdution and discussoin parts are sound and the relevant references are mentioned.

Overall, we believe the trial has substantial design problems which will be difficult to overcome. Furthermore, the sample size seems to small to allow correction for all significant cofounders and likely will not provide sufficient knowledge even if the above mentioned critique is addressed.

Author Response

Szypulska et al performed a crosssectional study correlating immunosuppression and wine consumption with steroid- and non-steroid sex hormones in stable kidney transplant recipients. The authors hypothesized that in analogy to published in vivo data Tacrolimus-based immunosuppression reduces sex hormone production and circulatory levels in this patient cohort. The study consists of 119 kidney transplant recipients of both sexes, with a relatively large proportion of males and females aged 45 years and older. They quantified antropometric values, allograft function and steorid- and non-steorid sex levels in the plasma. Overall the authors claim that in female patients on Tac-based immunosuppression had a significantly lower level of Testosteron, Prolactin and Cortisol. In Males, incerased levels of Testerosteron and FSH were reported. Furthermore, the claim that Wine consumption (dichtomous variable) increases Cortisol levels.  

Overall, while correlation of immunosuppression and social behaviour on reproductive function is attractive, we identify various conceptional, design and statistical problems in this manuscript

Dear Reviewer #1:

Thank you very much for your revision. We really appreciate all your comments
and constructive criticism. We have considered all your suggestions, and we have improved
the manuscript using red colour; we hope that these changes meet with your approval.

Reviewer #1:

Cofounders: the study is ill controlled for relevant clinical and pre-analytic co-founders, including permenstrual state of female participants, comedication (including hormonal replacement therapy), auotimmune disorders and number of rejections, previous immunosuppression before and after transplantation, (Cyclophosphamide, Steroids, Induction Therapy, Chemotherapy), dialysis time and number of kidney and other solid organ transplants, prevalence of primary amenorrhea, interaction of hormone levels and eGFR.

 (response)
We are very grateful for your comments and criticism, however, with all the respect we believe that due to the fact that our group is heterogeneous, we have conducted detailed and careful statistical analysis. We have performed statistical tests to compare different subgroups, including control group. We have taken into account sex, age etc. Additionally, our main aim of the current study was to compare hormonal levels in kidney-transplant recipients receiving Tac. Due to this fact we had to compare patients receiving Tac (treatments group) with patients whose regimens excludes Tac (control group). We couldn’t choose any patients without immunosuppressive drugs, as it would be out of the point of current study. There are no kidney-transplant recipients who do not use any of the immunosuppressive drugs. To meet your expectation we have carried out additional analysis of renal parameters (creatinine, GFR),
we also have broaden the demographical and biochemical data of our patients. We believe,
that it is impossible to find representative group of patients suffering the same basis disease, having the same hormonal replacement therapy or taking the same regimens, as all of these therapies are usually tailored for the particular patient, are very individual.
Additionally following your advices we have enlarged the demographical and biochemical data of patients by the percentage of patients suffering from diabetes mellitus, time of functioning of the graft, percentage of patients using a particular immunosuppressive drug (Patients and methods).
We hope that revised version of the manuscript will meet your approval.

 Pre-analysis: The study does not describe in the methods details about sampling: association with menstruation in premenopausal women, time of blood sampling,

(response)

We are grateful for this important advice, we have provided these information in current version of manuscript (Patients and methods). All the blood samples were collected in the morning (7.30. am. – 9.30.am), additionally premenopausal women were in follicular phase of menstrual cycle when donating blood for tests.

Reporting Bias: A significant part of the manuscript covers correlation of wine consumption on sex hormone levels, although the study uses wine consuption as a dichotomous variable. Clearly, self reporting alcohol consumption has limitations in such research and must be analysed as a continous variable.

(response)

We are grateful for your comment. We realize that wine/alcohol drinking may be problematic to be estimated correctly when the cohort is very large as the intake is self-reporting. As we do not want to blur our results and we are aware of drawing incorrect conclusions we have decided to delete the section regarding alcohol intake in the revised version of our manuscript. We hope it will meet with your approval.

Text and References: The introdution and discussoin parts are sound and the relevant references are mentioned.

(response)

We really appreciate your comment, we are grateful for that.

Overall, we believe the trial has substantial design problems which will be difficult to overcome. Furthermore, the sample size seems to small to allow correction for all significant cofounders and likely will not provide sufficient knowledge even if the above mentioned critique.

(response)

We are grateful for this comment too. Many of our patients were enrolled in current study but due to fact, that they are administered different immunosuppressive regimens, we have found it very difficult to collect more numerous groups. Still, based on available and modern literature we believe, that the groups are numerous enough to count the significant statistic for them, and the conclusions that are drawn from the analysis may be important for clinicians and therefore for their patients too, which seems to be on the greatest importance for us.

Reviewer 2 Report

The authors examined the effects of tacrolimus-based therapy and wine drinking on the hormone status of grafted patients. This article was an important result in transplantation clinical research. But, authors should edit tables and figures to fit the text. 

Author Response

The authors examined the effects of tacrolimus-based therapy and wine drinking on the hormone status of grafted patients. This article was an important result in transplantation clinical research. But, authors should edit tables and figures to fit the text. 

Dear Reviewer #2:

We are very grateful for your advices and constructive comments concerning our manuscript. We really appreciate all your comments. We have improved the manuscript hoping that these changes meet with your approval.

Reviewer 3 Report

In this study, the authors assessed the level of sex hormones in tacrolimus-treated kidney transplanted patients and controls. 119 Patients were prospectively included.

This study is well conducted and show interesting results.

However, I have some major concerns:

  • First, the immunosuppression in the control group is very heterogeneous in my opinion, and it makes difficult to interpret the results in this group. More importantly, patients with Cyclosporine A are included in the control group which may be questionable when comparing to another CNI.

  • I think that the alcohol analyses are inappropriate and should be removed: this study ‘s aim is to assess the impact of Tacrolimus on sex hormones profile. The impact of alcohol use is another question. Moreover, the study assessed only the wine influence and no other alcohol use (why?) and is based on self-declaration which may not be reproductible data.

  • Table 1. The authors should explain the choice to separate in groups < 45 years and > 45 years? We expect a comparison between TAC + and Tac- patients in table 1. How can we interpret the difference in the TAC+ versus TAC- for hormones if we don’t know for instance if the age is similar in the two groups? (that is of major importance since the authors showed that values of hormones are not similar between young and older recipients); Is the renal function similar between TAC+ and TAC- patients? (as we know that CKD is associated with hormonal dysfunction). Moreover some important data are lacking in the baseline description of the population such as: Duration of kidney transplantation, Immunosuppression used (N(%) under MMF, steroids, mTORi, induction treatment), diabetes mellitus , time on dialysis and with CKD, transplantation rank..

  • Is the period of woman cycle important to interpret sex hormones level? If yes, should it be mentioned for women recipients ?

Minor comments:

  • The authors should homogenize terms: Immunosuppressive drugs (ID) immunosuppressive therapy (IT)

  • Figure 1. We expect here p-value between groups.

  • Page 4 line 162. due to [neurotoxicity of CNI]. TAC trough level is measured to assess nephrotoxicity mostly and under exposition for the risk of rejection

  • T stand for Testosterone I presume but I could not find the abbreviation in the text

Author Response

In this study, the authors assessed the level of sex hormones in tacrolimus-treated kidney transplanted patients and controls. 119 Patients were prospectively included.

This study is well conducted and show interesting results.

However, I have some major concerns.

Dear Reviewer #3:

Thank you very much for your revision. We really appreciate constructive criticism and all your comments. We have considered all your suggestions, and we have improved the manuscript using red colour; we hope that these changes meet with your approval.

First, the immunosuppression in the control group is very heterogeneous in my opinion, and it makes difficult to interpret the results in this group. More importantly, patients with Cyclosporine A are included in the control group which may be questionable when comparing to another CNI.

 (response)

We are grateful for this constructive criticism. The immunosuppression in the control group indeed is heterogeneous. Therefore we have conducted detailed and careful statistical analysis. We have performed statistical tests to compare different subgroups, including control group taking into account sex, age etc. Additionally our main aim of the current study was to compare hormonal levels in kidney-transplant recipients receiving Tac. Due to this fact we had to compare patients receiving Tac (treatments group) with patients whose regimens  excludes Tac (control group). We couldn’t choose any patients without immunosuppressive drugs, as it would be out of the point of current study. There are no kidney-transplant recipients who do not use any of the immunosuppressive drugs. 

I think that the alcohol analyses are inappropriate and should be removed: this study ‘s aim is to assess the impact of Tacrolimus on sex hormones profile. The impact of alcohol use is another question. Moreover, the study assessed only the wine influence and no other alcohol use (why?) and is based on self-declaration which may not be reproductible data.

(response)

Dear Reviewer, thank you very much for constructive criticism. We have considered your comments, and we admit that perhaps our survey and idea was improperly design. Therefore, following your advice we have removed this data. We hope that ow our manuscript will be more clear for readers and it will meet your expectations.  

 Table 1. The authors should explain the choice to separate in groups < 45 years and > 45 years? We expect a comparison between TAC + and Tac- patients in table 1. How can we interpret the difference in the TAC+ versus TAC- for hormones if we don’t know for instance if the age is similar in the two groups? (that is of major importance since the authors showed that values of hormones are not similar between young and older recipients); Is the renal function similar between TAC+ and TAC- patients? (as we know that CKD is associated with hormonal dysfunction). Moreover some important data are lacking in the baseline description of the population such as: Duration of kidney transplantation, Immunosuppression used (N(%) under MMF, steroids, mTORi, induction treatment), diabetes mellitus , time on dialysis and with CKD, transplantation rank.

 (response)

Thank you for your comment. Physiologically menopause in our population is usually observed at the age of 50. Due to the fact that the appearance of menopause is diagnosed approximately 4.5 years earlier in transplant recipients, we took 45 as the cut-off point of age.  

Dear Reviewer, with all the respect, Table 1. contains demographical data, not the results concerning hormonal levels. We believed that summarizing all the demographical data in a form of table would be more clear and more attractive for readers. The comparison between the level of particular hormones are presented among the other ion Tables 2,3 and 4. Additionaly, Table 2 contains comparison between hormonal levels between sexes.

Following your comments and important advices, we have prepared additional demographical and biochemical data describing our groups of patients, and these data are presented in modified and enlarged Table 1.

According to your advice we have provided the comparison of renal function reflected by the comparison of GFR and creatinine level in group of  patients receiving Tac (TAC+) and excluding Tac (TAC-). Such comparison is presented in section Results and discussed in section Discussion.

Additionaly, we have assessed the median time of graft function of our patients as well as the number of patients suffering from diabetes mellitus. All the abovementioned data is presented in Table 1.

Following your comments, we have provided the graph presenting the percentage of patients taking a particular immunosuppressive drug.

Is the period of woman cycle important to interpret sex hormones level? If yes, should it be mentioned for women recipients ?

(response)

Thank you very much for this advice. Yes,  indeed it is very important in which phase was the woman when donating the blood. We are very grateful for this important comment. All the premenopausal women were in follicular phase of menstrual phase when donating the blood for the test. This information is now included in the revised version of manuscript in the section Patients and  methods.

Minor comments:

The authors should homogenize terms: Immunosuppressive drugs (ID) immunosuppressive therapy (IT).

(response)
Dear Reviewer, thank you for this comment, however with all the respect these two terms do not mean the same, therefore we cannot homogenize them. Immunosuppressive drug means a single drug, whereas the immunosuppressive therapy means a regimen containing at least two different immunosuppressive drugs.

Figure 1. We expect here p-value between groups.

(response)

We are grateful for this comment too, we are grateful for that. We have fixed this issue and the proper information is now available in revised version of the manuscript.

Page 4 line 162. due to [neurotoxicity of CNI]. TAC trough level is measured to assess nephrotoxicity mostly and under exposition for the risk of rejection

 (response)
We really appreciate your effort, thank you very much. We have fixed it in revised version of our manuscript.

T stand for Testosterone I presume but I could not find the abbreviation in the text

 (response)
Yes, indeed T stand for testosterone, thank you for this comment. We have added an abbreviation.

Round 2

Reviewer 1 Report

our requests for revisions were not met

Author Response

Dear Reviewer #1:

With all the respect, we have changed and improved our manuscript according to your previous comments and advices. We have broaden the demographical data of both groups of patients TAC+ and TAC- (control group), these data are presented in 2 separate tables (Tab1e 2). As can be seen from this comparison, the control group well reflects the characteristic of treatment group (TAC+), and actually the only important differentiating factor between these two groups is administration/not administration TAC.

Moreover, following your previous advices, we have described in more detail the blood sampling and the association between time of blood collecting and menstrual phase of female donors.

Additionally, we have deleted the section of our manuscript regarding wine consumption, as taking into account your comments and advices this part of our study was probably improperly designed.

We have also added and more widely discussed the issue of correlation between cyclosporin A and tacrolimus (Discussion), hoping that now our manuscript is more clear and available for readers.

We believe that our revised version of manuscript as well as our explanation will find your acceptation and meet with your approval.

Reviewer 3 Report

Thank you for the consideration of the comments.

The authors answers most of the remarks :

I must insist in two points :

  • First because of the methodology : as well said by the authors : the aim is to compare hormones between patients treated with TAC and those who are not. This objective is relevent. However, is this case, we NEED a more detailed and easily accessible baseline demographic comparison between TAC + and TAC- patients.
  • Second : we all agree that the control group is heterogeneous. it is important but this choice is understandable. However, I am very uncomfortable with keeping patients on cyclosporine. Tacrolimus and cyclosporine act on the same molecules and have almost the same side effects. I suspect that by removing them, there must not be many patients left in the control group. But this point needs to be discussed further. Comparison of hormonal effects between TAC and ciclo ?

Author Response

Thank you for the consideration of the comments. The authors answers most of the remarks.

Dear Reviewer #3:

Thank you very much for your revision. We really appreciate all your comments
and constructive criticism. We have considered your suggestions, and we have improved
the manuscript; we hope that these changes meet with your approval.

I must insist in two points :

  • First because of the methodology : as well said by the authors : the aim is to compare hormones between patients treated with TAC and those who are not. This objective is relevent. However, is this case, we NEED a more detailed and easily accessible baseline demographic comparison between TAC + and TAC- patients.

(response)

Thank you for the comment and advice. We have significantly broaden the demographical data of our patients. We have presented them in current version of the manuscript in 2 separate tables (Table 1,2) hoping that they are more attractive, clear and accessible.

  • Second : we all agree that the control group is heterogeneous. it is important but this choice is understandable. However, I am very uncomfortable with keeping patients on cyclosporine. Tacrolimus and cyclosporine act on the same molecules and have almost the same side effects. I suspect that by removing them, there must not be many patients left in the control group. But this point needs to be discussed further. Comparison of hormonal effects between TAC and ciclo ?

(response)

We appreciate your comment, thank you. We have completed and broaden the discussion section (Discussion) according to your advices. We have explained the association between tacrolimus and cyclosporine and we have emphasized that we examined the regimen (multidrug treatment), not the single immunosuppressive drug, since it better reflects the real condition of treatment of renal-transplant-recipients in current study and in clinical practice. Although TAC and CsA belong to the same groups of immunosuppressants, they are different drugs, with different properties, therefore they can act differently on hormonal status. Moreover, patients who are treated with CsA constitute only 15,3% of all patients while patients on TAC represent 74% of all.  The aim of current study was to assess the TAC+ regimens versus regimens without TAC. As there are no patients taking only one immunosuppressive drug in our study, all of the regimens represent multidrug therapy. We hope that revised version of our manuscript  and our explanations will meet your expectation and it will find your approval.